# Screening for Osteoporosis from Blood Test Data in Elderly Women Using a Machine Learning Approach

**DOI:** 10.3390/bioengineering10030277

**Published:** 2023-02-21

**Authors:** Atsuyuki Inui, Hanako Nishimoto, Yutaka Mifune, Tomoya Yoshikawa, Issei Shinohara, Takahiro Furukawa, Tatsuo Kato, Shuya Tanaka, Masaya Kusunose, Ryosuke Kuroda

**Affiliations:** 1Department of Orthopaedic Surgery, Kobe University Graduate School of Medicine, Kusunoki-cho, 7-5-1, Chuou-ku, Kobe City 650-0017, Japan; 2Orthopaedic Surgery Kobe Rosai Hospital, Kagoike-dori 4-1-23, Chuou-ku, Kobe City 651-0053, Japan

**Keywords:** machine learning, osteoporosis, artificial intelligence

## Abstract

The diagnosis of osteoporosis is made by measuring bone mineral density (BMD) using dual-energy X-ray absorptiometry (DXA). Machine learning, one of the artificial intelligence methods, was used to predict low BMD without using DXA in elderly women. Medical records from 2541 females who visited the osteoporosis clinic were used in this study. As hyperparameters for machine learning, patient age, body mass index (BMI), and blood test data were used. As machine learning models, logistic regression, decision tree, random forest, gradient boosting trees, and lightGBM were used. Each model was trained to classify and predict low-BMD patients. The model performance was compared using a confusion matrix. The accuracy of each trained model was 0.772 in logistic regression, 0.739 in the decision tree, 0.775 in the random forest, 0.800 in gradient boosting, and 0.834 in lightGBM. The area under the curve (AUC) was 0.595 in the decision tree, 0.673 in logistic regression, 0.699 in the random forest, 0.840 in gradient boosting, and 0.961, which was the highest, in the lightGBM model. Important features were BMI, age, and the number of platelets. Shapley additive explanation scores in the lightGBM model showed that BMI, age, and ALT were ranked as important features. Among several machine learning models, the lightGBM model showed the best performance in the present research.

## 1. Introduction

Osteoporosis is a systemic skeletal disorder characterized by low bone mass and the microarchitectural deterioration of bone tissue, which leads to bone fragility. Osteoporosis increases fracture risk in elderly populations, and it is the most common cause of vertebral fractures and femoral neck fractures among them. These fractures significantly reduce the activities of daily living. Osteoporosis commonly affects postmenopausal women due to decreased estrogen secretion [1]. Identifying patients at high risk of fracture prior to a fracture occurring is critical in osteoporosis care. The standard method to diagnose osteoporosis involves the estimation of bone mineral density (BMD) via dual-energy X-ray absorptiometry (DXA) in the proximal femur, lumbar spine, or forearm, and then BMD is compared with that of a reference group. According to the guideline from the Japan Osteoporosis Society, the diagnostic criteria for osteoporosis require a T-score of BMD less than −2.5 or less than 70% of the Young Adult Mean (YAM) [2]. In addition to the BMD measurement, the presence of fragility fractures, such as proximal hip or spinal fractures, is also diagnosed as osteoporosis [1,3].

Other risk factors for osteoporotic fractures are age, previous fragility fracture, the previous occurrence of hip fracture in a parent, and the use of corticosteroids [3]. The Fracture Risk Assessment Tool (FRAX) is a model for fracture risk assessment that uses 12 input parameters: age, sex, weight, height, previous fracture, parental fracture history, rheumatoid arthritis, use of glucocorticoids, secondary osteoporosis, smoking, and alcohol consumption [4]. Its output is the 10-year probability of fractures. One of FRAX’s limitations involves its modeling assumptions. Some parameters, such as smoking, are dichotomously used regardless of severity. Therefore, a more accurate and helpful tool for screening the risk of osteoporosis is required.

Artificial intelligence has recently become more popular in the medical field because of the tremendous progress in machine learning (ML) algorithms [5]. In fact, the U.S. Food and Drug Administration has authorized some healthcare companies to sell artificial intelligence technology to medical professionals [6]. The use of ML applications is rapidly expanding in various areas of health research, and ML has the potential to improve the current healthcare system and clinical practice. In 2015, Madelin et al. [7] compared several ML classifiers for diagnosing osteoarthritis of the knee joint using MRI data. They concluded that linear models had the best performance compared to other classifiers, such as neural networks, K-nearest neighbors (KNN), or support vector machines (SVM).

In the field of osteoporosis studies, ML has already been applied to predict hip fractures. In 2017, Kruse et al. [8] used a clustering model, an unsupervised ML method, to extract nine layers of different osteoporotic fracture risks from 10,775 subjects. A later study by the same group used twenty-four different models to identify hip fractures from a Danish cohort of 5439 women and men [9]. This study classified the cohort into nine groups depending on fracture risk. Some studies used imaging data to identify prevalent vertebral fractures. By combining global and local density and texture parameters of non-fractured vertebral regions in computed tomography (CT) scans, the ML model outperformed BMD alone in detecting the presence of vertebral fractures [10]. This ML model is helpful for screening individuals with a high risk of osteoporotic fractures. In 2017, Shoji et al. [11] used a neural network as an ML model. They concluded that predicting future bone density values using artificial neural networks was valid for tailor-made medical care. In 2022, Huang et al. [12] used abdominal computed tomography images of the psoas muscle by extracting the features using a radiomics approach followed by an ML algorithm. Gradient Boosting Machine showed the best performance in their study.

Developing efficient screening and diagnostic tools for osteoporosis may have clinical impacts since it will improve patients’ prognoses through early disease detection. The gold standard for an osteoporosis diagnosis is the use of DXA. However, this method requires specific equipment and is unavailable in some clinics. The use of DXA also involves radiation exposure. We hypothesized that the ML approach could predict whether the YAM value of the patient was lower than 70% or not without taking X-rays or CT scans. This study used the ML approach to predict low BMD without measuring DXA. We used patient age, body mass index, and blood test data from routine blood examinations for ML model training to achieve this aim. Several ML algorithms were compared based on the prediction performance for the test data. Additionally, important parameters for outcome prediction were determined using the ML technique.

## 2. Materials and Methods

### 2.1. Ethical Approval

This study was approved by the Institutional Review Board (IRB) of Kobe Rosai Hospital (approval number 30-01).

### 2.2. Data Collection

The medical records of 2541 women (average age 73.5 ± 9.2 years old) who visited an outpatient clinic for the diagnosis and treatment of osteoporosis were used in this study. As hyperparameters for machine learning, patient age, body mass index (BMI), and blood test data were used (Table 1). The data used from blood tests were white blood cells (WBCs: counts/µL), hemoglobin (Hb: g/dL), platelets (Plt: countsx104/µL), total protein (TP: g/dL), albumin (Alb: g/dL), aspartate transferase (AST: IU/L), alanine transaminase (ALT: IU/L), gamma-glutamyl transpeptidase (gammaGTP: IU/L), alkaline phosphatase (ALP: IU/L), calcium (Ca: mg/dL), creatine kinase (CK: IU/L), chloride (Cl: mEq/L), sodium (Na: mEq/L), potassium (K: mEq/L), magnesium (Mg: mg/dL), creatinine (Cr: mg/dL), blood urea nitrogen (BUN: mg/dL), uric acid (UA: mg/dL), tartrate-resistant acid phosphatase 5b (TRACP5b: mU/dL), Bone-Specific Alkaline Phosphatase (BAP: µg/L), Procollagen I Intact N-Terminal (PINP: ng/mL), and Estimated Glomerular Filtration Rate (eGFR: mL/min). The patient group was classified into two groups (osteoporosis or non-osteoporosis) according to the value of the femur neck YAM.

### 2.3. Machine Learning

Five ML algorithms, logistic regression, decision tree, random forest, gradient boosting trees, and lightGBM, which is a modified version of gradient boosting trees, were used to compare the model performance (Figure 1). All ML algorithms were implemented using Scikit-learn, a free ML library for Python [13]. To create the training data, the patient data were randomly divided into a training sample (80%) and a test sample (20%). The test sample was used to test the performance of each model. Logistic regression models are widely used in medical research for multivariate analysis. Decision tree analysis is a schematic representation of several decisions followed by different chances of occurrence. Three algorithms, random forest, gradient boosting, and lightGBM, are common tree-based ensemble methods known to be the most accurate models on various datasets [14]. These models combine multiple simple tree models to produce reliable predictions. Training data were used to determine the optimal hyperparameters for each algorithm, and each model prediction was evaluated using test data. From the confusion matrix, accuracy, recall, precision, and f-measure were calculated. The area under the curve (AUC) from the receiver operating characteristic (ROC) was also calculated. The 95% confidence interval (CI) for each endpoint was computed using the bootstrap method, where 300 replacement resampling procedures were performed to evaluate the model performance. The bootstrap method is an iterative replacement resampling technique that is used in ML studies to estimate the summary statistics [14,15]. The results were statistically analyzed using R Studio (RStudio PBC, Boston, MA, USA). Data were expressed as the mean and standard deviation. Statistical significance was denoted by *p* < 0.001. Two different algorithms were used to visualize each predictive parameter’s important values. Permutation feature importance is defined as the decrease in a model’s score when a single feature value is randomly shuffled. The Shapley additive explanation (SHAP) value is defined as the contribution of each feature to the model prediction based on game theory. All analyses were conducted using Python (version 3.8) and Scikit-learn (version 1.0.2).

## 3. Results

### 3.1. Study Participants

Of the total 2541 patients, 799 were diagnosed with osteoporosis based on femoral YAM values and 1742 were non-osteoporotic. The data used for machine learning are summarized in Table 1. The data of the osteoporosis group and the normal group were compared using an unpaired *t*-test. Age, Hb, Plt, Alb, ALT, ALP, CK, Cl, Na, Cr, BUN, TRACP5b, PINP, eGFR, and BMI showed statistically significant differences between the two groups (Table 2). The correlation coefficients between the femur YAM and each parameter are summarized in Table 3.

### 3.2. Prediction of Osteoporosis in each Machine Learning Model

The predictive results of each model are summarized in Table 4. Among five machine learning models, lightGBM showed the highest scores for accuracy, precision, and f-measure. The score for recall was the highest in the random forest model. The hyperparameters used in model training are summarized in Table 5. Figure 2 shows the ROC curves of each algorithm. AUC was 0.595 in the decision tree, 0.673 in the logistic regression, 0.699 in the random forest, 0.840 in gradient boosting, and 0.961, which was the highest, in the lightGBM model.

## 3.3. Importance Values of the Predictors

To detect the important parameters for the diagnosis of osteoporosis, the feature importance was calculated in the tree-based model. As a result, BMI, age, and the number of platelets were ranked as the three most important parameters for the prediction of osteoporosis in the lightGBM model (Figure 3a). The SHAP scores showed that BMI, age, and ALT were ranked as important feature values. The SHAP values of BMI and ALT positively correlated with the femur YAM, while age was negatively correlated with the femur YAM (Figure 3b).

## 4. Discussion

The Osteoporosis Self-Assessment Tool (OST) is the oldest and simplest known method of identifying osteoporotic patients using patient weight and age data [16,17]. Another simple screening tool for the prediction of future fracture risk is FRAX. FRAX requires 12 parameters, such as demographic characteristics, lifestyle, and past medical history. However, the recent literature has highlighted the limitations of these approaches [18]. In recent decades, ML models, an AI-based approach, have been increasingly used to predict osteoporosis. In 2021, Wang et al. [19] compared seven machine learning models to detect patients with or without osteoporosis. In 2021, Ou-Yang et al. [20] implemented five ML models using 16 to 19 features (physical findings and blood tests) to identify the risk of osteoporosis in the population of Taiwan. Trained ML models in the study showed significantly better performance compared to the traditional OST model. In a study using patient age, height, weight, and blood test data of Vietnamese women, seven different machine learning models were compared [18]. The random forest model showed the best AUC value on test data (AUC = 0.81). Age, height, and weight were shown as important features of the ML model.

The present study used patient age, BMI, and 22 parameters from blood test data. The parameters of blood test data were routinely used in regular clinical situations and covered by health insurance. Comparing five different models—logistic regression, decision tree, gradient boosting, random forest, and lightGBM—lightGBM showed the best scores in accuracy, recall, f-measure, and AUC. LightGBM is known as a gradient boosting framework that uses decision-tree-based ensemble algorithms [21]. Among the tree-based boosting family of algorithms, the lightGBM model employs a histogram algorithm and a depth-limited leaf-wise growth strategy to increase computational efficiency, reduce memory usage, improve classification accuracy, and efficiently prevent overfitting. LightGBM has been used for the prediction of neurological prognoses after cervical spinal cord injuries [22]. The algorithm has been used to detect early stages of ovarian cancer or bladder cancer using blood test and tumor marker data [23,24].

The interpretation of the model performance is important in medical AI research since a clinician is responsible for making a reasonable decision based on AI prediction. This logic is called XAI, which represents explainable AI. The goal of XAI is to create models that are more easily explained, understood, and effectively managed by humans while maintaining high forecasting accuracy. As XAI tools, two methods were used in the present study. The permutation feature importance is defined as the amount by which the model score decreases when one feature is randomly shuffled. This procedure breaks the relationship between the feature and the target. In the present study, BMI, the number of platelets, and age were important parameters. SHAP is another XAI approach that is used to explain a machine learning model’s prediction by calculating the contribution of each feature to the prediction based on game theory. BMI, age, and ALT showed high SHAP scores in the present model. The SHAP values of BMI and ALT were positively correlated with the femur YAM, while age was negatively correlated with the femur YAM. It is clinically reasonable that patient age and height and weight or BMI data were important factors in screening for osteoporosis. The results are similar to previous reports on machine learning and osteoporosis [18,20]. In the present study, platelet count and ALT were also important factors in the study model; Table 2 shows that these factors had statistically significant differences between groups. However, Table 3 shows that these factors do not correlate strongly with the femoral YAM and are not clinically recognized as risk factors for osteoporosis. In 2015, Breitling [25] reported that liver enzymes were weakly correlated with osteoporosis in the general population. Regarding the number of platelets, an article from Korea reported that platelet counts are significantly associated with osteopenia and osteoporosis in middle-aged and elderly people [26]. We speculate that blood cells are created in the bone marrow and may indirectly represent bone turnover or quality.

There are several limitations in the study. Firstly, the model showed good performance in the present dataset. However, the population of the data is based on an outpatient clinic for osteoporosis treatment in an urban area of Japan. Data from different populations, such as rural areas, where people are engaged in agriculture or fishing, might have different results. Secondly, we used only female records in this study since we did not have enough records of male cases. Although osteoporosis commonly affects postmenopausal women, it is also recognized as a clinical issue among older men [1]. A prediction model that can be used in males and females will be needed. Thirdly, we did not use imaging data such as X-rays or CT scans. There are several studies that used imaging data and neural networks to improve the model performance [10,12]. Developing a model combined with imaging data will be the next step of this study. Finally, we used XAI tools for model explanations to understand the importance of each parameter. However, XAI tools, like statistical models, are difficult to explain clearly with regard to how they work. It is hoped that better XAI tools will emerge in the future.

## 5. Conclusions

In conclusion, the machine learning approach was used to predict low-BMD patients using patient age, body mass index (BMI), and blood test data. A trained ML model could predict whether the BMD was less than 70% or not with high probability. Among five ML models, the lightGBM model showed the best performance in the present research.

## Figures and Tables

**Figure 1 bioengineering-10-00277-f001:**
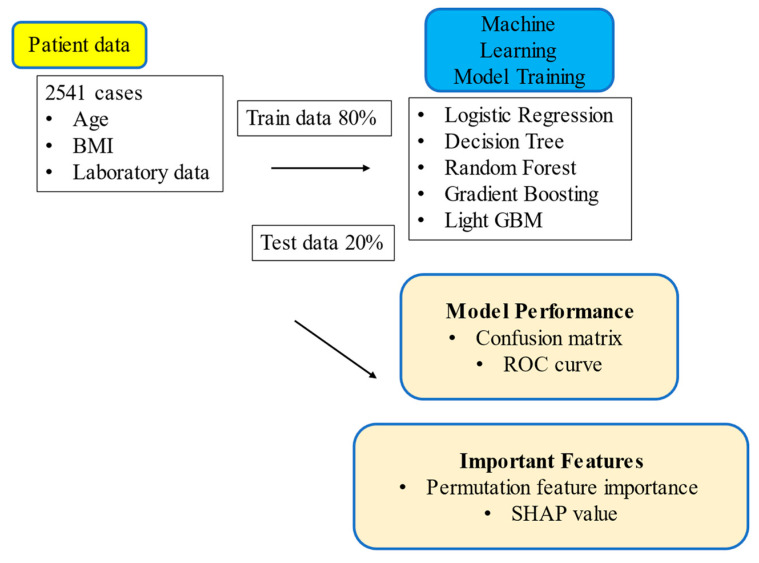
Workflow of data collection and machine learning. The data were divided into training data and test data, and five machine learning models were used. The trained models were evaluated using test data, and important features were visualized.

**Figure 2 bioengineering-10-00277-f002:**
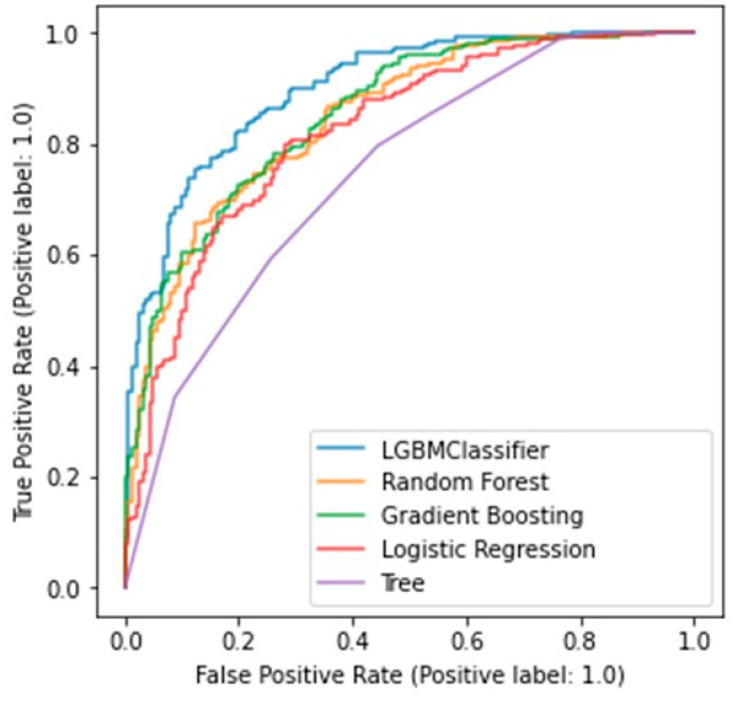
ROC curve of each trained model (LGBMClassifier: lightGBM; Tree: decision tree).

**Figure 3 bioengineering-10-00277-f003:**
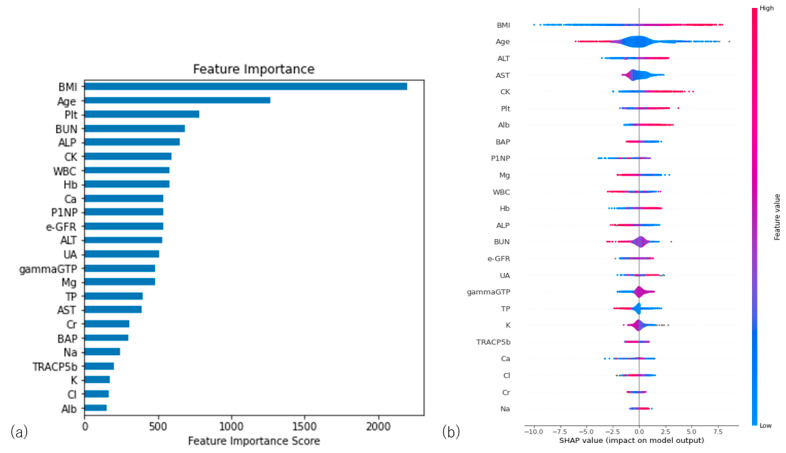
(**a**) Permutation feature importance of lightGBM model. Important features have more significant scores. The top three important features were BMI, age, and the number of platelets. (**b**) SHAP value of lightGBM model. The top three important features were BMI, age, and ALT. The warm color shows a positive impact on model performance, while the cool color shows a negative impact.

**Table 1 bioengineering-10-00277-t001:** Mean ± SD of patient age, blood test data, and BMI.

Data for Machine Learning (Abbreviation)	Mean	SD
Age	73.5	9.1
White blood cells (WBCs: counts/µL)	5783.5	1485.5
Hemoglobin (Hb: g/dL)	12.7	1.2
Platelets (Plt: counts × 10^4^/µL)	22.7	6.1
Total protein (TP: g/dL)	7.1	0.4
Albumin (Alb: g/dL)	4.1	0.3
Aspartate transferase (AST: IU/L)	22.9	7.6
Alanine transaminase (ALT: IU/L)	17.0	9.9
Gamma-glutamyl transpeptidase (gammaGTP: IU/L)	24.3	27.8
Alkaline phosphatase (ALP: IU/L)	222.2	87.1
Calcium (Ca: mg/dL)	9.4	0.4
Creatine kinase (CK: IU/L)	115.4	83.2
Chloride (Cl: mEq/L)	105.1	2.6
Sodium (Na: mEq/L)	141.1	2.2
Potassium (K: mEq/L)	4.1	0.3
Magnesium (Mg: mg/dL)	2.1	0.2
Creatinine (Cr: mg/dL)	0.7	0.4
Blood urea nitrogen (BUN: mg/dL)	17.8	5.8
Uric acid (UA: mg/dL)	4.8	1.3
Tartrate-resistant acid phosphatase 5b (TRACP5b: mU/dL)	379.6	172.0
Bone-Specific Alkaline Phosphatase (BAP: µg/L)	12.9	5.3
Procollagen I Intact N-Terminal (PINP: ng/mL)	49.6	37.9
Estimated Glomerular Filtration Rate (eGFR: mL/min)	66.5	17.6
Body mass index (BMI: kg/m^2^)	21.7	3.4

**Table 2 bioengineering-10-00277-t002:** Mean ± SD of each group, and *p* values of the unpaired t-test are shown in the right row.

	Osteoporosis Group (n = 799) (femur YAM < 70%)	Normal Group (n = 1742) (femur YAM ≥ 70%)	*p* Value
Data for Machine Learning (Abbreviation)	Mean ± SD	Mean ± SD	
Age	76.5 ± 8.5	72.1 ± 9.2	9.2 × 10^−30^
White blood cells (WBCs: counts/µL)	5703 ± 1571	5851 ± 1442	0.06
Hemoglobin (Hb: g/dL)	12.3 ± 1.2	12.8 ± 1.2	3.9 × 10^−28^
Platelets (Plt: counts × 10^4^/µL)	21.0 ± 5.7	23.1 ± 6.3	0.0001
Total protein (TP: g/dL)	7.1 ± 0.5	7.0 ± 1.4	0.02
Albumin (Alb: g/dL)	4.0 ± 0.3	4.1 ± 0.3	1.5 × 10^−11^
Aspartate transferase (AST: IU/L)	23.1 ± 7.6	22.8 ± 7.5	0.26
Alanine transaminase (ALT: IU/L)	15.4 ± 10.3	17.8 ± 9.7	1.8 × 10^−8^
Gamma-glutamyl transpeptidase (gammaGTP: IU/L)	23.4 ± 27.0	24.7 ± 28.1	0.26
Alkaline phosphatase (ALP: IU/L)	233.2 ± 108.7	217.1 ± 74.6	1.37 × 10^−5^
Calcium (Ca: mg/dL)	9.4 ± 0.5	9.4 ± 0.4	0.11
Creatine kinase (CK: IU/L)	105.9 ± 79.8	119.9 ± 84.5	8.6 × 10^−5^
Chloride (Cl: mEq/L)	104.8 ± 2.9	105.2 ± 2.5	1.1 × 10^−3^
Sodium (Na: mEq/L)	140.8 ± 2.3	141.2 ± 2.2	8.7 × 10^−5^
Potassium (K: mEq/L)	4.1 ± 0.4	4.1 ± 0.3	0.54
Magnesium (Mg: mg/dL)	2.1 ± 0.2	2.1 ± 0.2	0.02
Creatinine (Cr: mg/dL)	0.8 ± 0.6	0.7 ± 0.3	5.3 × 10^−7^
Blood urea nitrogen (BUN: mg/dL)	18.9 ± 6.4	17.3 ± 5.4	1.5 × 10^−5^
Uric acid (UA: mg/dL)	4.8 ± 1.4	4.8 ± 1.3	0.71
Tartrate-resistant acid phosphatase 5b (TRACP5b: mU/dL)	399.1 ± 204.1	370.6 ± 154.4	1.0 × 10^−5^
Bone-Specific Alkaline Phosphatase (BAP: µg/L)	13.3 ± 6.8	12.7 ± 4.4	0.004
Procollagen I Intact N-Terminal (PINP: ng/mL)	53.3 ± 44.2	47.9 ± 34.4	6.7 × 10^−5^
Estimated Glomerular Filtration Rate (eGFR: mL/min)	64.0 ± 19.2	67.6 ± 16.7	2.1 × 10^−6^
Body mass index (BMI: kg/m^2^)	20.1 ± 2.9	22.5 ± 3.3	3.9 × 10^−66^

**Table 3 bioengineering-10-00277-t003:** The correlation coefficients between femur YAM > 70% and parameters for machine learning.

Data for Machine Learning (Abbreviation)	Correlation Coefficient between Femur YAM
Age	−0.22
Blood urea nitrogen (BUN: mg/dL)	−0.13
Creatinine (Cr: mg/dL)	−0.1
Alkaline phosphatase (ALP: IU/L)	−0.09
Tartrate-resistant acid phosphatase 5b (TRACP5b: mU/dL)	−0.09
Procollagen I Intact N-Terminal (PINP: ng/mL)	−0.08
Bone-Specific Alkaline Phosphatase (BAP: µg/L)	−0.07
Magnesium (Mg: mg/dL)	−0.05
Total Protein (TP: g/dL)	−0.04
Aspartate Transferase (AST: IU/L)	−0.02
Potassium (K: mEq/L)	−0.01
Uric acid (UA: mg/dL)	−0.01
gamma-glutamyl transpeptidase (gammaGTP: IU/L)	0.02
Calcium (Ca: mg/dL)	0.03
White blood cells (WBCs: counts/µL)	0.04
Platelets (Plt: counts × 10^4^/µL)	0.08
Creatine kinase (CK: IU/L)	0.08
Chloride (Cl: mEq/L)	0.08
Sodium (Na: mEq/L)	0.08
Estimated Glomerular Filtration Rate (eGFR: mL/min)	0.09
Alanine transaminase (ALT: IU/L)	0.11
Albumin (Alb: g/dL)	0.13
Hemoglobin (Hb: g/dL)	0.22
Body mass index (BMI: kg/m^2^)	0.34

**Table 4 bioengineering-10-00277-t004:** Summary of the predictive results of each model.

ML Model	Logistic Regression (95% CI)	Decision Tree (95% CI)	Random Forest (95% CI)	Gradient Boosting (95% CI)	Light GBM (95% CI)
Accuracy	0.772 (0.768–0.776)	0.739 (0.735–0.743)	0.775 (0.769–0.781)	0.800 (0.794–0.806)	0.834 (0.827–0.841)
Precision	0.771 (0.768–0.774)	0.737 (0.734–0.742)	0.764 (0.760–0.769)	0.800 (0.795–0.801)	0.835 (0.827–0.843)
Recall	0.956 (0.952–0.960)	0.968 (0.966–0.970)	0.978 (0.976–0.980)	0.957 (0.954–0.959)	0.961 (0.959–0.963)
F-measure	0.853 (0.851–0.856)	0.837 (0.834–0.840)	0.858 (0.855–0.862)	0.870 (0.867–0.874)	0.891 (0.887–0.896)

**Table 5 bioengineering-10-00277-t005:** Summary of hyperparameters of each model. (lbfgs: limited-memory Broyden–Fletcher–Goldfarb–Shanno; auc: area under the curve).

ML Model	Logistic Regression	Decision Tree	Random Forest	Gradient Boosting	Light GBM
Representative parameters	Penalty: l2 C: 100 Solver: lbfgs	Criterion: gini Max depth: 3	Max depth: 6 Number of estimators: 300 Scoring: auc	Learning rate: 0.19 Number of estimators: 100 scoring: auc	Number of itertions: 1000 Max depth: 6 Scoring: auc

## Data Availability

The data presented in this study are available upon request from the corresponding author. The data are not publicly available because of confidentiality concerns.

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
