# Peer review of "Screening for Osteoporosis from Blood Test Data in Elderly Women Using a Machine Learning Approach"

_bioengineering, 2023, doi:10.3390/bioengineering10030277_

Round 1

Reviewer 1 Report

The paper presents a possible quantification of osteoporosis evolution as a function of laboratory data based on machine learning protocols. The text is reasonably well written and method sound. Results are clearly presented. Updates and clarifications are required prior to considering possible publication.

- The exact sens of the two words “form” in the title is unclear as per the overall meaning of the title. The authors are suggested to revise this title form in order to make it directly understandable.

- Abstract : “...using DXA in an elderly woman … a total of 2540 females … was used…” → did the authors use one or 2540 subjects ?

- Abstract and other places in paper : “… laboratory data…” → The sens of this text comes clear when reaching Table 1 or Table 2, but before that it could mean anything. Please make sure the reader knows that the study is based on “patient biology” and nothing else.

- Abstract : “...70%…” → this also comes clear later on the text as it is described what it is, but it is not directly obvious in the abstract section.

- Abstract : “...this method can be used to screen…” → this affirmation is based on the 0.834 score of LightGBM method, but it is not said that this value is sufficient enough. Please rephrase or argue.

- Intro, line 36 : “...since or less…” → not very clear. The word since seems correct, but does not go with less.

- Intro, line 41 : Please define FRAX at first occurrence

- Intro, line 55, 56 : When using references with names, please do not put the ref number at the end of sentence, but just behind the names. Also check all places in the paper where this occurs.

- Intro, line 64 : repeat ML.

- Intro, line 73 : It seems it should be Algorithm.

- M&M, line 88 : Ethical approval → Institute, which one ?

- M&M, line 94 : See previous comment, please define “Laboratory data”. Also, appears “Table 1” but the caption is not defined next to the Table …

- Results, line 129 : Please rephrase “Out of a total of 2541 case,…”

- Results, line 129 : “799 cases were classified…” → the word classified is not clear and not methodologically defined

- Results, line 130 : “The features of the hyperparameters…” → please define features (as what they mean)

- Results, line 134, 135 : “According to … the patient” → This sentence is not abvious when look at the heat map in terms of colors. Please clarify this point.

- It seems Table 3 caption is not defined, please check

- Results, line 150, 151 : “As a result, BMI … age were…” → Looking at Figure 4a, it does not seem to be the first 3 features by order of importance, can you explain this choice ? Also, when looking at Figure 4b, similar argument pertains … please explain.

- Figure 4 : It feels somewhat not comfortable, not to have same parameters in line of Fig 4a and Fig 4b directly in front so it is kind of easy to compare the two together.

- Discussion, line 196, 197 : “...and enable human users can understand, …” → not properly written, please adjust.

- Discussion, line 205-207 : “BMI and number of ...with the presence of osteoporosis” → As earlier in the text this argument is put forward, but does not seem to be obviously correlated to clear evidence in the results extracted. Please extract this evidence in order to make this statement obvious as a result.

Author Response

Reviewer1

The paper presents a possible quantification of osteoporosis evolution as a function of laboratory data based on machine learning protocols. The text is reasonably well written and method sound. Results are clearly presented. Updates and clarifications are required prior to considering possible publication.

>Thank you for reviewing our manuscript. We made a point-by-point response. 

- The exact sense of the two words “form” in the title is unclear as per the overall meaning of the title. The authors are suggested to revise this title form in order to make it directly understandable.

>Thank you for the suggestion.I realized the misspelling. We corrected the grammatical error of the title.

- Abstract : “...using DXA in an elderly woman … a total of 2540 females … was used…” did the authors use one or 2540 subjects ?

>Data of 2540 subjects were used. We corrected the grammar error.

- Abstract and other places in paper : “… laboratory data…” The sense of this text comes clear when reaching Table 1 or Table 2, but before that it could mean anything. Please make sure the reader knows that the study is based on “patient biology” and nothing else.

> Thank you for the suggestion. The word ‘laboratory data’ was changed to ‘blood test data’.

- Abstract : “...70%…” → this also comes clear later on the text as it is described what it is, but it is not directly obvious in the abstract section.

>We rephrased the sentence. ‘Each model was trained to classify to predict low BMD patients.’

- Abstract : “...this method can be used to screen…” → this affirmation is based on the 0.834 score of LightGBM method, but it is not said that this value is sufficient enough. Please rephrase or argue.

> Thank you for the comment. The phrase is overstated. We deleted the sentence from the abstract.

- Intro, line 36 : “...since or less…” → not very clear. The word since seems correct, but does not go with less.

>We rephrased the sentence. ‘diagnostic criteria of osteoporosis require the T-score of BMD less than -2.5 or less than 70% of Young Adult Mean (YAM)’

- Intro, line 41 : Please define FRAX at first occurrence

>We defined FRAX as the following. ‘Fracture Risk Assessment Tool(FRAX) is a ,,,’

- Intro, line 55, 56 : When using references with names, please do not put the ref number at the end of sentence, but just behind the names. Also check all places in the paper where this occurs.

> Thank you for the comment. We check all references.

- Intro, line 64 : repeat ML.

> We rephrased the sentences, L63-66.

- Intro, line 73 : It seems it should be Algorithm.

> We corrected the grammatical error.

- M&M, line 88 : Ethical approval → Institute, which one ?

> We changed the sentence. ‘This study was approved by the Institutional Review Board (IRB) of Kobe Rosai Hospital.’

- M&M, line 94 : See previous comment, please define “Laboratory data”. Also, appears “Table 1” but the caption is not defined next to the Table …

>The word ‘laboratory data’ was changed to ‘blood test data’. We added the caption of Table 1.

- Results, line 129 : Please rephrase “Out of a total of 2541 case,…”

>We rephrased it as ‘Of total 2541 cases’

- Results, line 129 : “799 cases were classified…” → the word classified is not clear and not methodologically defined.

>We rephrased as follows, ‘Of the total 2541 patients, 799 were diagnosed with osteoporosis based on femoral YAM values and 1742 with non-osteoporosis.’.

- Results, line 130 : “The features of the hyperparameters…” please define features (as what they mean)

>We rephrased the sentence, ‘The data used for machine learning are summarized in Table 2’.

- Results, line 134, 135 : “According to … the patient”  This sentence is not obvious when look at the heat map in terms of colors. Please clarify this point.

> Since the heatmap was confusing, we changed the colormap of the heatmap. We also added Table 3 which summarized the correlation between each parameter and YAM. 

- It seems Table 3 caption is not defined, please check

>We added the caption.

- Results, line 150, 151 : “As a result, BMI … age were…” Looking at Figure 4a, it does not seem to be the first 3 features by order of importance, can you explain this choice ? Also, when

looking at Figure 4b, similar argument pertains … please explain.                                                                                            

> We corrected the error and rephrased the sentence. ‘As a result, BMI, age, and number of platelets were ranked as the three most important parameters for prediction of osteoporosis in light GBM model (Figure 4a). SHAP score showed BMI, age and ALT were ranked as important feature values. SHAP value of BMI and ALT positively correlated with femur YAM, while age was negatively correlated with femur YAM (Figure 4b).’

- Figure 4 : It feels somewhat not comfortable, not to have same parameters in line of Fig 4a and Fig 4b directly in front so it is kind of easy to compare the two together.

> I realized the figure 4b was the previous version of our experiment and replaced it with correct SHAP figure. Also, I corrected the code error of SHAP plotting method so that all the parameters appear on the Figure 4b. –

 Discussion, line 196, 197 : “...and enable human users can understand, …” → not properly written, please adjust.

> We corrected the phrase. ‘The goal of XAI is to create models that are more easily explained, understood and effectively managed by humans, while maintaining high forecasting accuracy.

- Discussion, line 205-207 : “BMI and number of ...with the presence of osteoporosis” → As earlier in the text this argument is put forward, but does not seem to be obviously correlated to clear evidence in the results extracted. Please extract this evidence in order to make this statement obvious as a result.

>We added Table 3 to explain the correlation between the parameters. Also, we added some sentences in discussion. ‘In the present study, platelet count and ALT were also important factors in the study model; Table 2 shows that these factors had statistically significant differences between groups. However, Table 3 shows that these factors do not correlate strongly with femoral YAM and are not clinically recognized as risk factors for osteoporosis.’

Reviewer 2 Report

Specific points and suggestions for improvement of the manuscript are listed below.

General comments:
(1) When ever you cite in text, please add year of publication.

(2) The authors had 2540 cases (L91) but in the results they report 2541 cases (L129).

(3) Can the authors give a bit more details about their models. How many epochs they were running and what was the loss function.

(4) The heatmap in figure 2 is very confusing. I expected zero to be the interphase between cool and warm colors but it’s closer to 0.3. When I first read the results (“According to the heatmap, YAM value positively correlated with BMI and negatively correlated with the age of the patient”; L134-5) I thought it was a mistake (because both BMI and age are blue (cool color)). I had to read the image 2 more times before I got it. I think it make more sense to have all positive correlations in one color scheme and all negative correlations in another color scheme.

(5) Tree-based model (figure 4a) and SHAP (figure 4b) are not showing the same parameters. SHAP is missing BUN, WBC, Ca, Cr, K, Cl. The tree-based model is missing RBC and Ht. Even stranger, RBC and Ht do not appear in the original blood data (table 2) or the heatmap (figure 2). So where did the authors get these data from and why weren’t them part of the analysis against femur YAM?

(6) The authors found that BMI and age have strong correlation with YAM BMD of the femur. This is not surprising and was shown in many studies before. However, these are not sensitive enough to predict osteoporosis (many older people with low BMI have normal femur BMD). The other 2 parameters (# of RBC and Plt) are a black box and the authors can’t really explain why they are important. Their explanation about bone turnover (L212-3) is unconvincing seeing that other more bone-related parameters (e.g. Ca and BAP) did not show any strong correlation.

Specific comments:

- L30: “… fracture femoral neck fracture among them”. should be “fracture and femoral neck fracture”.

- L50: “Madelin et al. compared”. Please add year of publication. (please correct all other instances of missing year).

- Table 1: Title/legend is missing. Furthermore, please add units to parameters.

- Table 3: Title/legend is missing.

- Figure 2: seeing that the 2 halves across the diagonal are a mirror-image it makes more sense just to show 1-side to not confuse the reader. For that matter, since the authors only care about the correlation with “femur YAM”, why not show that row. Maybe also add the actual correlation value in each square.

Author Response

Reviewer 2

General comments:

  • Whenever you cite in text, please add year of publication.

>We added the publication year to the text.

(2) The authors had 2540 cases (L91) but in the results they

report 2541 cases (L129).

> We corrected the miscounting. The total number of case is 2541.

(3) Can the authors give a bit more details about their models.

How many epochs they were running and what was the loss

function.

>We added the details (representative model parameters) in Table 4.

(4) The heatmap in figure 2 is very confusing. I expected zero to be the interphase between cool and warm colors but it’s closer to 0.3. When I first read the results (“According to the heatmap, YAM value positively correlated with BMI and negatively correlated with the age of the patient”; L134-5) I thought it was a mistake (because both BMI and age are blue (cool color)). I had to read the image 2 more times before I got it. I think it make more sense to have all positive correlations in one color scheme and all negative correlations in another color scheme.

>Thank you for your comment. I agree that the heatmap was confusing. Therefore we changed the color bar of the heatmap. Also, the label ‘femur YAM’ was replaced with ‘femurYAM>70%’.

(5) Tree-based model (figure 4a) and SHAP (figure 4b) are not showing the same parameters. SHAP is missing BUN, WBC, Ca, Cr, K, Cl. The tree-based model is missing RBC and Ht. Even stranger, RBC and Ht do not appear in the original blood data (table 2) or the heatmap (figure 2). So where did the authors get these data from and why weren’t them part of the analysis against femur YAM?

> I realized the figure 4b was the previous version of our experiment and replaced it with correct SHAP figure. Also, I corrected the code error of SHAP plotting method so that all the parameters appear on the Figure 4b.

(6) The authors found that BMI and age have strong correlation with YAM BMD of the femur. This is not surprising and was shown in many studies before. However, these are not sensitive

enough to predict osteoporosis (many older people with low BMI have normal femur BMD). The other 2 parameters (# of RBC and Plt) are a black box and the authors can’t really explain why they are important. Their explanation about bone turnover (L212- 3) is unconvincing seeing that other more bone-related parameters (e.g. Ca and BAP) did not show any strong

correlation.

>Thank you for the comment, as described in the discussion some machine learning model has a black box even though an explainable AI method is used. Unpaired t- test in table 2 showed some parameters (including platelet and ALT) have a statistically significant difference between osteoporosis and the normal group. We added some description in the discussion part as well as limitation part.   

‘In 2015, Breitling25 reported that liver enzyme weakly correlated with the presence of osteoporosis in general population. Regarding the number of platelets, an article from Korea reported that platelet counts are significantly associated with osteopenia and osteoporosis in middle-aged and elderly people26.’ ‘Finally, we used XAI tools for model explanation to understand the importance for each parameter. However, XAI tools, like statistical models, are difficult to explain clearly how they work. It is hoped that better XAI tools will emerge in the future.’

Specific comments:

- L30: “… fracture femoral neck fracture among them”. should be “fracture and femoral neck fracture”.

>We corrected the sentence.

- L50: “Madelin et al. compared”. Please add year of publication. (please correct all other instances of missing year).

>We corrected the reference citation.

- Table 1: Title/legend is missing. Furthermore, please add units to parameters.

- Table 3: Title/legend is missing.

>We corrected the table data,

- Figure 2: seeing that the 2 halves across the diagonal are a mirror-image it makes more sense just to show 1-side to not confuse the reader. For that matter, since the authors only care

about the correlation with “femur YAM”, why not show that row. Maybe also add the actual correlation value in each squart.

>We changed the appearance of the heatmap. Since we hoped to show a correlation between other factors, we showed all the values. We added a new table which shows the correlation between femurYAM and other parameters.

Round 2

Reviewer 2 Report

The authors revised the manuscript and the new version is much improved. Yet I still struggle with the heatmap. I compared the previous version to the new one and I found several differences I can’t explain. The authors state that “we changed the color bar of the heatmap”, yet when looking at the color bar I can see that the range was also changed (from originally (-0.4 to 1.0) to (-0.7 to 1.0)). Furthermore, some correlations changed. For example, in the original heatmap, the correlation between e-GFR and BUN (dark blue) was weaker than the correlation between e-GFR and Cr (light blue). In the revised heatmap, the correlation between e-GFR and BUN (purple) is stronger than the correlation between e-GFR and Cr (dark purple). Similarly, in the original heatmap, the correlation between PINP and ALP (light read) was stronger than the correlation between BAP and ALP (light green). In the revised heatmap, the correlation between PINP and ALP (purple) is weaker than the correlation between BAP and ALP (dark green). There are other similar changes but these 2 are enough. Changing the color of the heatmap and even changing the range should not change the relative correlation between parameters. Can the authors explain what happened?

Author Response

Dear reviewer

Thank you for reviewing our manuscript in detail. We corrected the manuscript and commented on your review. Please see the attachment.

Reviewer’s comment

The authors revised the manuscript, and the new version is much improved. Yet I still struggle with the heatmap. I compared the previous version to the new one and I found several differences I can’t explain. The authors state that “we changed the color bar of the heatmap”, yet when looking at the color bar I can see that the range was also changed (from originally (-0.4 to 1.0) to (-0.7 to 1.0)). Furthermore, some correlations changed. For example, in the original heatmap, the correlation between e-GFR and BUN (dark blue) was weaker than the correlation between e-GFR and Cr (light blue). In the revised heatmap, the correlation between e-GFR and BUN (purple) is stronger than the correlation between e-GFR and Cr (dark purple). Similarly, in the original heatmap, the correlation between PINP and ALP (light read) was stronger than the correlation between BAP and ALP (light green). In the revised heatmap, the correlation between PINP and ALP (purple) is weaker than the correlation between BAP and ALP (dark green). There are other similar changes but these 2 are enough. Changing the color of the heatmap and even changing the range should not change the relative correlation between parameters. Can the authors explain what happened?

>>Thank you for checking the heatmap. I realized two reasons caused the confusion.

First, some coding errors in plotting the heatmap. The scale for minimum and maximum correlation coefficient was not set correctly, which caused confusion on the color scale bar appearance. When we set the color bar scale correctly, the heatmap appearance will be the following figure.

[Figure- Please see the attachment.]

Another point is that while we created table 3 on the first round review, we realized there is an outlier value in some parameters ( ex. BMI was 3300, P1NP was 1300, and eGFR was 960; these values do not exist clinically, and caused by typing error when we create the 1st dataset ). Outliers did not significantly affect the machine learning model's performance, but clinically improbable values were manually removed, and the correlation coefficient was calculated.

However, the heatmap data is still confusing. Since we already have table 3, we removed the heatmap figure from our revised manuscript. 

Round 3

Reviewer 2 Report

Thank you for your replay. The submitted version is improved.